# Optimising Vaccine Dose in Inoculation against SARS-CoV-2, a Multi-Factor Optimisation Modelling Study to Maximise Vaccine Safety and Efficacy

**DOI:** 10.3390/vaccines9020078

**Published:** 2021-01-22

**Authors:** John Benest, Sophie Rhodes, Matthew Quaife, Thomas G. Evans, Richard G. White

**Affiliations:** 1Department of Infectious Disease Epidemiology, London School of Hygiene and Tropical Medicine, Keppel Street, London WC1E 7HT, UK; sophie.rhodes@lshtm.ac.uk (S.R.); matthew.quaife@lshtm.ac.uk (M.Q.); Richard.White@lshtm.ac.uk (R.G.W.); 2Vaccitech Ltd., The Schrodinger Building, Heatley Road, The Oxford Science Park, Oxford OX4 4GE, UK; tom.evans@vaccitech.co.uk

**Keywords:** dosing, dose-response, adenovirus-vectored vaccines, dose dynamics, COVID-19

## Abstract

Developing a vaccine against the global pandemic SARS-CoV-2 is a critical area of active research. Modelling can be used to identify optimal vaccine dosing; maximising vaccine efficacy and safety and minimising cost. We calibrated statistical models to published dose-dependent seroconversion and adverse event data of a recombinant adenovirus type-5 (Ad5) SARS-CoV-2 vaccine given at doses 5.0 × 10^10^, 1.0 × 10^11^ and 1.5 × 10^11^ viral particles. We estimated the optimal dose for three objectives, finding: (A) the minimum dose that may induce herd immunity, (B) the dose that maximises immunogenicity and safety and (C) the dose that maximises immunogenicity and safety whilst minimising cost. Results suggest optimal dose [95% confidence interval] in viral particles per person was (A) 1.3 × 10^11^ [0.8–7.9 × 10^11^], (B) 1.5 × 10^11^ [0.3–5.0 × 10^11^] and (C) 1.1 × 10^11^ [0.2–1.5 × 10^11^]. Optimal dose exceeded 5.0 × 10^10^ viral particles only if the cost of delivery exceeded £0.65 or cost per 10^11^ viral particles was less than £6.23. Optimal dose may differ depending on the objectives of developers and policy-makers, but further research is required to improve the accuracy of optimal-dose estimates.

## 1. Introduction

Severe Acute Respiratory Syndrome Coronavirus 2 (SARS-CoV-2) has been an unprecedented burden on global health throughout 2020 [1]. Due to the health risks associated with infection, countries have had to implement policies of isolation and quarantine, causing global disruption to economic and health systems [2]. Vaccination is a vital tool in improving global health [3] and an effective vaccine against SARS-Cov-2 could drastically reduce the spread of this highly infectious pathogen. There is an urgent need to accelerate the development of a SARS-CoV-2 vaccine to protect the population [4]. However, maintaining safety and immunogenicity standards within vaccine development is paramount. Decisions relating to vaccine development need to be made quickly and accurately. One important decision is determining vaccine dose, defined as the quantity or magnitude of vaccine given.

Model-based drug development is commonly used to accelerate drug decision making, and the field of Immunostimulation/Immunodynamic (IS/ID) modelling has been developed to adapt these methods for vaccine development [5]. IS/ID modelling has shown promise in discovering optimal dose for TB and influenza inoculations [6,7] and for exploring dose-response trends for adenoviral-vectored vaccines [8], however, the previous works have focused entirely on optimising dose with respect to immunological response. In these studies, the modelling of dose-response and hence finding the dose that maximises response can be considered single-objective optimisation problems. 

Whilst optimising response with respect to vaccine dose is essential to ensuring effective vaccination, the change in financial cost and safety of vaccination with respect to vaccine dose are also important. A vaccine should ideally maximise protective immunogenicity, minimise the risk of vaccine-related toxicity and minimise the cost of using that vaccine. Optimising dose in relation to immunogenicity, safety and cost is a multi-objective or multi-criteria-decision-analysis problem [9]. Using models to analyse the dose-response, dose-safety and dose-cost relationships can provide insight into the multi-dimensional dose-utility curve and hence optimal dose. There may also exist cases where the cost of vaccination can be ignored, for example, if the vaccine is not limited in supply. This could also be true for cases where there is a very high willingness to pay for vaccination by policymakers, which permits costs to be nearly ignored to ensure a maximum reduction in disease burden.

Approximately 180 SARS-CoV-2 vaccines are in development [10], and there is much interest in which vaccines may offer the greatest efficacy in reducing the burden of SARS-CoV-2 on global health [11,12,13]. Not only should the dose of any vaccine implemented be optimised with regards to safety, immunogenicity and cost, but appropriate dosing is equally important to ensure that unbiased estimate of vaccine immunogenicity and safety is used. In addition to establishing optimal dose at an individual level, the potential of a candidate vaccine to induce herd immunity in an entirely vaccinated population can also be considered. 

Whilst accelerating these decisions and improving dosing is important now [14], it is also hoped that modelling and multi-objective optimisation can aid in rapid vaccine response to future epidemics. 

In this work, we aimed to optimise the dose of a recombinant adenovirus type-5 (Ad5) vectored SARS-CoV-2 vaccine using IS/ID modelling and multifactorial optimisation. Our objectives were:(1)Using published data, calibrate mathematical models to the relationship between dose and seroconversion, safety and cost of a single inoculation.(2)Identify the minimum dose that is predicted to theoretically induce herd immunity.(3)Identify the dose that maximises immunogenicity and safety.(4)Identify the dose that maximises immunogenicity and safety whilst minimising cost.

## 2. Materials and Methods

### 2.1. Data

Data were extracted from a published phase 1 study of a SARS-Cov-2 adenoviral-based vaccine [15]. Data on neutralising antibody-based seroconversion and adverse events were extracted for 108 healthy human participants inoculated with the candidate vaccine on day 0. Individuals were divided into three dose groups (5 × 10^10^, 1 × 10^11^ and 1.5 × 10^11^ Viral Particles (VP)) and their responses were measured at day 28. 

Seroconversion was chosen as a surrogate of protective immunity and was defined as a neutralising antibody response post-vaccination of at least a four-fold increase relative to baseline, above which an individual was protected [16]. 

Safety was defined by the proportion of individuals that experienced adverse events within 0–28 days post-vaccination. Vaccine adverse events can be graded depending on severity (Table 1), with grade 3 or above considered severe [17,18]. As a measure of safety, we considered both the proportion that experienced “any grade” adverse events and the proportion that experienced grade 3 or above adverse events. Whilst the designation “Grade 3+” is used to be consistent with the terms used by the original authors, no adverse events above grade 3 were reported.

### 2.2. Objective 1. Using Published Data, Calibrate Mathematical Models to the Relationship between Dose and Seroconversion, Safety and Cost of a Single Inoculation

#### 2.2.1. Dose-Seroconversion Relationship

We calibrated a sigmoid function to the dose-seroconversion data using the nls function in R [19,20]. Sigmoid functions are commonly used to describe biological processes, including in both drug and vaccine response modelling. The formula is
(1)$$sigmoid\left(Dose\right)=BaseResponse+\frac{MaxResponse-BaseResponse}{1+e^{-Scale\times \left(Dose-Dose_{50}\right)}}$$
where *Dose* was log10 (Viral Particles), and *BaseResponse* (the minimum output of the function), *MaxResponse* (the maximum output of the function), *Dose*50 (the dose which defines the functions midpoint) and *Scale* (which determines the steepness of the curve) were the model parameters. 

We also calibrated a representative non-saturating (“peaking”) curve, using the methods discussed in [8]. However, as these methods did not support the non-saturating curve providing a better description of the data, we assume that the dose-response follows a sigmoidal function (Appendix A).

In the absence of seroconversion data from a placebo group, a base seroconversion rate of 0% was assumed (*BaseResponse* = 0). We predicted the dose-seroconversion curve for doses of up to 10^15^ VP, to ensure previous adenoviral dosing ranges are explored [21] and state the dose that would induce 50% and 90% seroconversion. 

#### 2.2.2. Dose-Safety Relationship

We calibrated a sigmoid function (Equation (1)) to the dose—”any grade adverse event” data using the nls function in R, and another to the dose—”grade 3+ adverse event” data. Again, in the absence of a placebo group, a base adverse event rate of 0% was assumed for both curves (*BaseResponse* = 0). Further, we assumed that for sufficiently large doses, 100% of individuals would experience adverse events. We predict the probability of an adverse event for doses of up to 10^15^ VP.

Lastly, we found the doses for which the proportion of individuals that would experience grade 3+ adverse events above the thresholds of 30% and 17%. 30% of grade 3+ adverse events has been defined as a threshold for unacceptable toxicity in dose-escalation studies [22,23,24]. However, of the commonly CDC recommended vaccines, the largest grade 3+ adverse reaction rate is 17% for the Shingrix herpes zoster vaccine [25].

#### 2.2.3. Dose-Cost Relationship

We assumed that the cost of a single individual receiving a vaccination can be described by
(2)$$Cost_{Total}\left(Dose\right)=Cost_{Delivery}+Cost_{Dose-dependant}\left(Dose\right)$$
where $Cost_{Total}\left(Dose\right)$ was the total cost in British Pounds Sterling (GBP, £) to vaccinate one person with a given dose. We assumed that this was the sum of “*Delivery*” costs, which are independent of dose, and the dose-dependent cost. Expected costs for doses of up to 10^15^ VP were calculated using this formula and parameters in Table 2.

Specifically, the delivery costs included the cost of disposable materials used in vaccination (gloves, sterilizing alcohol, prefilled syringes, needles), storage (assumed to be 1 month of storage costs), and personnel costs (15 min of nursing time). The delivery cost formula was therefore calculated as
(3)$$Cost_{Delivery}\left(Dose\right)=Cost_{Materials}+Cost_{Storage}+Cost_{Personnel}$$
where
(4)$$Cost_{Materials}\left(Dose\right)=Cost_{Gloves}+Cost_{Alcohol}+Cost_{PFS}+Cost_{Needles}$$
(5)$$Cost_{Storage}\left(Dose\right)=Cost_{Storagepermonth}\times 1\left(months\right)$$
(6)$$Cost_{Personnel}\left(Dose\right)=\left(AnnualWage÷AnnualHours\right)\times Timepervaccination$$

The dose-dependent cost was the cost of the manufactured adenoviral vaccine measured in viral particles, assuming bulk production, which increased linearly with the vaccine dose. The dose-dependent cost formula was therefore calculated as
(7)$$Cost_{Dose-dependent}=Viralparticlespervaccination\left(Dose\right)\times Costperviralparticle$$

With
(8)$$Costperviralparticle=Cost_{AdenoviralBatch}÷\left(AdenoviralConcentration\times BatchVolume\right)$$

These costs were a simplification of real-world costs but represent an approximate cost of vaccination. The delivery cost of vaccination (*Cost_Delivery_*) was calculated as £5.24, and the cost per 10^11^ VP (*Cost_Dose-dependent_*) of adenovirus was £0.76 ( Table 2). 

### 2.3. Objective 2. Identify the Minimum Dose that Is Predicted to Theoretically Induce Herd Immunity

An optimal vaccine dose should maximise response (maximise the proportion that will seroconvert), safety (minimise the proportion that experience adverse events), and affordability (minimise cost per vaccination). Increasing dose may increase seroconversion, but would also increase cost and adverse event prevalence. Vaccinating a population is often done to induce herd immunity, and so a factor in selecting the optimal dose is whether that dose could induce herd immunity in an entirely vaccinated population. Therefore, one approach to dose-optimisation is to choose a dose that can induce herd immunity within the population, and specifically to choose the minimum such dose to minimise cost and adverse event prevalence. 

We used the suggested 65.5% of the population required to be protected to cause herd immunity in the United Kingdom (UK) [33] to establish whether there exist doses for this vaccine that could induce this 65.5% seroconversion and hence induce herd immunity in an entirely vaccinated population. We then identified the minimum such dose that could do so. A 95% confidence interval for optimal dose was determined using a parametric bootstrapping approach (Appendix A).

### 2.4. Objective 3. Identify the Dose that Maximises Immunogenicity and Safety

Another approach to dose optimisation is to choose the dose which maximises the proportion of individuals that “safely” seroconvert. To do this, a multifactorial utility function was derived, defined here as a mathematical formula that estimates the “worth” or utility of doses relative to each other.

Using the assumption that the probability of seroconversion, *P_s_*, and the probability of grade 3+ adverse events, *P_t_*, were mutually independent, the probability of a safe seroconversion was equal to
(9)$$P_{S}\times \left(1-P_{t}\right)$$

(Figure 1). Therefore, the utility function, *U_Costless_*, to be maximised was
(10)$$U_{Costless}\left(Dose\right)=P_{s}\left(Dose\right)\times \left(1-P_{t}\left(Dose\right)\right)$$

Optimal dose was defined as the dose that maximised this function. A 95% confidence interval for optimal dose was determined using a parametric bootstrapping approach (Appendix A).

### 2.5. Objective 4. Identify the Dose that Maximises Immunogenicity and Safety Whilst Minimising Cost

We can also include the increased cost associated with an increased dose into the previous utility function (Equation (10)). In the case where cost is included as a potential limiting factor, a potential costed utility function was
(11)$$U_{Costed}\left(Dose\right)=\frac{P_{s}\left(Dose\right)\times \left(1-Pt\left(Dose\right)\right)}{Cost_{Total}\left(Dose\right)}$$

Using this utility function, we predicted the optimised dose for maximising seroconversion and minimising adverse events and cost. Note that *U_Costless_(Dose)* (Equation (10)) is precisely the numerator of the *U_Costed_(Dose)* (Equation (11)). A 95% confidence interval for optimal dose was again determined using a parametric bootstrapping approach (Appendix A).

#### Threshold Analysis 

Due to the difficulty in accurately estimating cost parameters, we conducted a threshold analysis on the parameter values of the cost model (Equations (3)–(8)). This was conducted to determine how much error would be needed in our costing model parameters to qualitatively alter the optimal predicted dose. We chose 5 × 10^11^, 1 × 10^11^ and 5 × 10^10^ VP as the thresholds of interest. 

To conduct a threshold analysis of parameters *Cost_Delivery_*, we fixed all other parameters at the calibrated/literature derived value and allowed *Cost_Delivery_* to vary. The region over which we varied *Cost_Delivery_* was +/− 3 orders of magnitudes of the value (£5.24) we used in the main model. In other words, we considered the effect of *Cost_Delivery_* being 1000 times larger or smaller (from £0.0052 per vaccination to £5240 per vaccination) on the prediction of optimal dose. This range was considered certainly to almost contain a reasonable estimate of the dose-independent costs of a single vaccination. This procedure was then repeated for *Cost per 10^11^* viral particles, ranging from £0.00076 to £760 per 10^11^ VP.

We found the parameter ranges for which the dose that optimised *U_Costed_(Dose)* were above and below the stated thresholds (5 × 10^11^, 1 × 10^11^, 5 × 10^10^ VP).

## 3. Results

### 3.1. Objective 1. Using Published Data, Calibrate Mathematical Models to the Relationship between Dose and Seroconversion, Safety, and Cost of a Single Inoculation

#### 3.1.1. Does-Seroconversion Relationship

The empirical data showed that doses of 5.0 × 10^10^, 1.0 × 10^11^ and 1.5 × 10^11^ induced 50%, 50%, 75% seroconversion on day 28, respectively. The calibrated saturating dose-seroconversion curve is displayed in Figure 2a. 50% and 95% seroconversion were predicted at a dose of 5.9 × 10^10^ and 2.4 × 10^12^ VP, respectively. Population demographics including age, gender and pre-existing adenovirus neutralising antibody titre were described [15] (Appendix A). 

#### 3.1.2. Dose-Safety Relationship

The study showed that doses of 5 × 10^10^, 1 × 10^11^ and 1.5 × 10^11^ VP induced 86%, 83% and 75% any grade adverse events and 6%, 6% and 17% grade 3+ adverse events, respectively. The calibrated saturating dose-adverse event curves are displayed in Figure 2b,c. The two thresholds of safety we previously chose were 17% and 30% grade 3+ adverse reaction proportion. The calibrated dose-adverse curve predicted that a rate of adverse events greater than 17% occurs for doses in excess of 1.58 × 10^11^ VP and exceeds 30% at 2.45 × 10^11^ VP. 

### 3.2. Objective 2. Identify the Minimum Dose that Is Predicted to Theoretically Induce Herd Immunity

The dose-seroconversion prediction for the minimum dose that could induce theoretical herd immunity is shown in Figure 3a. Given that an estimate for the critical herd immunity threshold in the UK has been estimated as 65.5%, a dose of 1.3 × 10^11^ VP would be required to reach this threshold, assuming the entire UK population was vaccinated. The 95% confidence interval for optimal dose was (8.0 × 10^10^, 7.9 × 10^11^) (Appendix A). Using the dose-safety model, this dose was predicted to cause 13.5% of vaccinated individuals to have a grade 3+ adverse event. 

### 3.3. Objective 3. Identify the Dose that Maximises Immunogenicity and Safety

The dose-utility prediction is shown in Figure 3b. The dose that optimised this function was 1.5 × 10^11^ VP (Figure 3b, red diamond). It was predicted that dosing at this magnitude would lead to a seroconversion rate of 67.6%, and cause 15.8% of vaccinated individuals to have a grade 3+ adverse event (83.0% any grade adverse events). 

Sensitivity analysis (Appendix A) showed that the prediction of the optimal dose was most sensitive to variance in the $Dose_{50}$ parameter of the dose-seroconversion sigmoid function and the $Dose_{50}$ parameter of the dose-safety sigmoid function (Appendix A). These were respectively equal to the doses that were predicted to induce 50% of vaccinated individuals to seroconvert and 50% to experience grade 3+ adverse events (with an increase in these parameters qualitatively shifting the curves in Figure 2a,c to the right). The 95% confidence interval for optimal dose was (2.9 × 10^10^, 5.0 × 10^11^) (Appendix A).

### 3.4. Objective 4. Identify the Dose that Maximises Immunogenicity and Safety Whilst Minimising Cost

The dose-utility relation including cost is shown in Figure 3c. The dose that optimised this function was 1.1 × 10^11^ VP. It was predicted that dosing at this magnitude would lead to a seroconversion rate of 62.20%, cost £6.07 per dose, and cause 10.32% of vaccinated individuals to have a grade 3+ adverse event (82.2% any grade adverse events). The 1 × 10^11^ VP dose had the highest utility of the doses tested in the study, and both of the 5 × 10^10^ and 1.5 × 10^11^ VP doses appeared to be near-optimal. This analysis, therefore, suggested that if the cost was included in the utility function then a marginally reduced dose was found optimal relative to the costless utility function. The predicted cost is within the expected range [$5–$37] for a single SARS-CoV-2 vaccine dose [34].

Again, we found that the prediction of the optimal dose was most sensitive to variance in the $Dose_{50}$ parameter of the dose-seroconversion sigmoid function. This parameter is equal to the dose that we predict would induce 50% of vaccinated individuals to seroconvert (with an increase in these parameters qualitatively shifting the curve in Figure 2a to the right). Optimal dose was not sensitive to <10% error in the estimation of $Cost_{Delivery}$ or Costperviralparticle (Appendix A). The 95% confidence interval for optimal dose was (2.1 × 10^10^, 1.5 × 10^11^) (Appendix A).

#### Threshold Analysis 

For *Cost_Delivery_*, we found that the predicted optimal dose was independent of the parameter value for large values (Figure 4). We found that the optimal dose was in excess of 1 × 10^11^ and 5 × 10^10^ VP for *Cost_Delivery_* values in excess of £3.79 and £0.65, respectively (hence optimal dose was only less than 5 × 10^10^ VP for *Cost_Delivery_* less than £0.65). These values were respectively 0.7 and 0.1 times the value that was used in the main analysis. We find that the optimal dose was not in excess of 5 × 10^11^ VP for any *Cost_Delivery_* values.

For *Cost per 10^11^ viral particles_,_* we found that the optimal dose was independent of the parameter value for large values (Figure 5). We found that the optimal dose was in excess of 1 × 10^11^ and 5 × 10^10^ VP for *Cost per 10^11^ viral particles* values in less than £1.06 and £6.23, respectively (hence optimal dose was only less than 5 × 10^10^ VP for *Cost per 10^11^ viral particles* greater than £6.24). These values were respectively 1.3 and 8.2 times the value that was used in the main analysis. We find that the optimal dose was not in excess of 5 × 10^11^ VP for any *Cost per 10^11^ viral particles* values.

We additionally explored varying both parameters simultaneously (Appendix A). If *Cost per 10^11^ viral particles* was less than 0.2 times *Cost_Delivery_*, then the predicted optimal dose was between 1.0 × 10^11^ and 1.5 × 10^11^ VP.

## 4. Discussion

Vaccination is an important part of global healthcare and disease prevention. Vaccination must be protective, safe, and affordable at a population level, and all of these factors may be impacted by dose. We used modelling and multifactorial optimisation approaches to predict the optimal dose of an adenoviral vectored vaccine against SARS-CoV-2 based on protection, safety and cost. A dose of 1.1 × 10^11^ VP of this vaccine was found to be optimal with respect to seroconversion, safety and cost. However, an increased dose of 1.3 × 10^11^ VP or 1.5 × 10^11^ VP could be justified depending on the objectives of developers and policymakers. These methods highlight how quantitative analysis can be used to ensure that vaccines are dosed optimally, and could aid in accelerating vaccine development.

The IS/ID methods used in this work have previously been used to analyse and optimise vaccine dose-response [7,34]. Compared to those studies, this work is novel in its inclusion of dose-safety and dose-cost models and multi-objective optimisation methods. This work used data published as part of a vaccine development protocol, which highlights how these methods do not require additional complexity in trial design. Using only the published data, we were able to hypothesize the best dosing for a candidate SARS-CoV-2 vaccine. Such methods could routinely be used to evaluate dose for other clinical and preclinical vaccine trials. Given the pandemic setting in which the SARS-CoV-2 candidates are being developed where trials are being accelerated and progressing faster than would normally be expected, modelling may be even more of an important adjunct in ensuring optimal vaccine dosing. 

We had to make some assumptions in this work. Firstly, the assumed cost function was based on a simplification of a vaccine campaign cost estimate suggested by the World Health Organisation [35,36], discounting costs incurred by vaccine wastage and incremental costs of maintaining hospitals and clinics. Additionally, the exact cost per viral particle of the vaccine was unknown and would vary with production scale, however, the threshold analyses showed that the optimal-dose prediction may still be robust despite this uncertainty. As this was a financial rather than economic analysis, we did not account for “Disability Adjusted Life Years” [37] (DALYs) gained by reducing SARS-CoV-2 impact or societal costs incurred by missing work to be vaccinated. Including DALYs or similar measures into the cost function may give a greater understanding of whether this vaccine would be a cost-effective approach to controlling SARS-CoV-2. However, the focus of the work was not in producing a fully functional economic evaluation of implementing this vaccine. Additionally, to analyse potential DALYs averted would have required the inclusion of economic and epidemiological models that were beyond the scope of this paper. 

An additional assumption of the utility model was that avoiding grade 3+ adverse events was as important to the utility function as inducing seroconversion. To address this, a weighting function may be applied if the expected discomfort of a SARS-CoV-2 infection is greatly in excess of a grade 3+ adverse event and a discomfort ratio between these two outcomes can then be determined (see Appendix A). Alternatively, thresholds for acceptable levels of seroconversion and adverse events could be determined, and any doses predicted to meet these thresholds considered optimal.

We found that dosing at 2.45 × 10^11^ VP would likely induce grade 3+ adverse events in greater than 30% of individuals vaccinated, which is a typical threshold for safety in clinical trials. Previous work has found that human-hosted adenoviral vector vaccine trials typically do not dose in excess of 2 × 10^11^ [21]. This suggests that adenoviral vaccine trials are being dosed at magnitudes that ensure that grade 3+ adverse reactions remain below the 30% threshold. However, for this vaccine, the available data was not sufficient to determine whether the dose-seroconversion curve shape was better described by a peaking or saturating curve shape. This implies that we cannot be confident that the percentage of individuals that seroconvert would continue to increase as dose increases beyond those empirically tested. This is likely the result of using too few doses or not dosing at a sufficiently large dose to observe peaking or saturating dose-response behaviour. We have previously shown that curve shape could not be determined for 75% of adenoviral dose-response data [8]. However, in this case, it is possible that dosing at a large enough magnitude to determine curve shape could cause an unacceptable number of grade 3+ adverse events. 

There were limiting factors to our analyses. We had to assume that seroconversion implies that an individual was protected against SARS-CoV-2 infection. This seemed appropriate in the absence of a validated model for predicting SARS-CoV-2 protective immunity or a challenge study, but could be updated as a greater understanding of SARS-CoV-2 correlates of protection is developed [38,39,40]. Additionally, we had to assume that the base seroconversion and adverse events percentage was 0%. That is to say that individuals that received no vaccine dose would not seroconvert or experience any adverse events. This was reasonable given the lack of a placebo group in the data but may limit the predictive validity of toxicity and seroconversion at lower doses. 

Further limitations are due to the non-inclusion of potential population effects and covariates in the model. The proportion of individuals that the vaccine needs to protect may change depending on the number of individuals that have been previously infected or on the extent that a prior infection provides lasting immunity. Additionally, prior adenoviral exposure or age of vaccinated individuals could impact the probabilities of seroconversion and adverse events. Individuals younger than 45 were shown to be less likely to experience fever and more likely to experience seroconversion [15] which, given that there existed some heterogeneity in age distribution between dosing groups in the data, may impact the model’s future predictive validity. Finally, given that no grade 4 (serious/life-threatening) events were observed, no analyses could be done to assess the dose-varying probability of these events.

This work implies that the doses that have been trialled for this vaccine were near the theoretical optimal dose. Whilst we predicted 1.1 × 10^11^ VP of the vaccine would be the dose that optimises safety, cost, and protective immunity, if vial size restricts precision on which doses can be administered then, of the previously empirically tested doses, both the 1.0 × 10^11^ and 1.5 × 10^11^ doses could be reasonable. We also predicted that inducing complete herd immunity in an otherwise entirely susceptible population may be feasible with this vaccine given 100% uptake, but we predict that approximately 13.5% of vaccinated individuals would experience grade 3+ adverse events and that this would require a dose of at least 1.3 × 10^11^ VP. As the dose optimising the costless utility function was in excess of this threshold, but not the dose optimising the utility function with cost, this work implies that to fully protect the UK population with this vaccine would require accepting some level of cost inefficiency. 

We anticipate the following future work. Firstly, to ensure that the models used to make these suggestions are accurate and valid, further clinical trials would need to be conducted, preferably at a wider spread of doses for which empirical data do not yet exist. In particular, a placebo group could allow for adjustment of the assumption that individuals that receive no vaccine dose do not seroconvert or experience any adverse events. This may help to increase accuracy in the prediction of optimal dose. Secondly, the simple assumptions made in developing the optimisation utility functions mean the function could be applied broadly but should be adapted to the specific knowledge and needs of vaccine developers and policymakers. For example, including a specific adverse reaction threshold that has been defined in the study protocol, or by predicting protective immunity as highlighted in [41]. 

Thirdly, these methods applied to other candidate SARS-CoV-2 vaccines may provide a method to compare the relative utility of these candidates. Fourthly, with respect to the dose-safety model, including a weighting of the relative risks/discomforts associated with SARS-CoV-2 infection/adverse events would be informative. Fifthly, the data was gathered from individuals residing in the Wuhan region only. Similar data should be gathered from other populations to assess potential differences in response and safety. Given sufficient data, incorporating the covariates of age and pre-existing rAd5 neutralising antibody titre into the model could aid in predictive validity across various populations and in assessing dose effect. 

Sixthly, this work only considers a single dose of the vaccine and response at one time-point. Further modelling could be attempted to address dose-optimisation the different time points or to consider a prime-boost paradigm. Finally, the dose-seroconversion and dose-safety models developed in this work could also be incorporated into the epidemiologic transmission and economic models to more accurately determine the health and economic impact of a given dose.

## 5. Conclusions

The SARS-CoV-2 pandemic has caused global health and economic issues and has led to increased pressure to rapidly develop a potentially life-saving vaccine. Dose is a key attribute in determining vaccine immunogenicity, safety and cost, and therefore dose-optimisation is an important aspect of vaccine development. Modelling and multifactorial optimisation methods allow for fast, quantitatively-based dosing decisions. Given the increased pressures for rapid vaccine development in response to pandemics, these tools should be considered a useful approach to accelerating vaccine development. 

## Figures and Tables

**Figure 1 vaccines-09-00078-f001:**
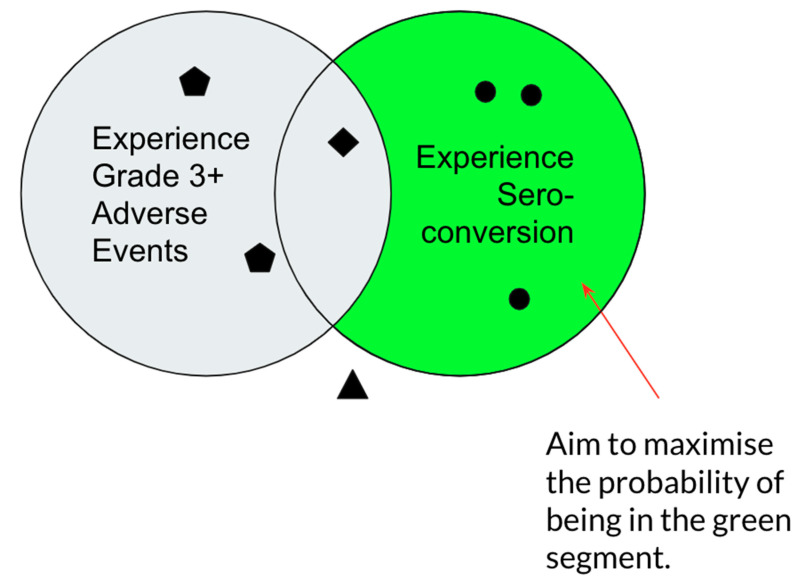
Venn diagram representation of possible outcomes of inoculation, where the left set includes individuals that experience grade 3+ adverse events and the right set includes individuals that experience seroconversion. We aimed to maximise the number of individuals that experience seroconversion and do not experience grade 3+ adverse events, represented in the green segment of the diagram. Black diamonds represent individuals that experience both outcomes, black pentagons represent individuals that experience grade 3+ adverse events with no seroconversion, and black triangles represent individuals that experience neither outcome.

**Figure 2 vaccines-09-00078-f002:**
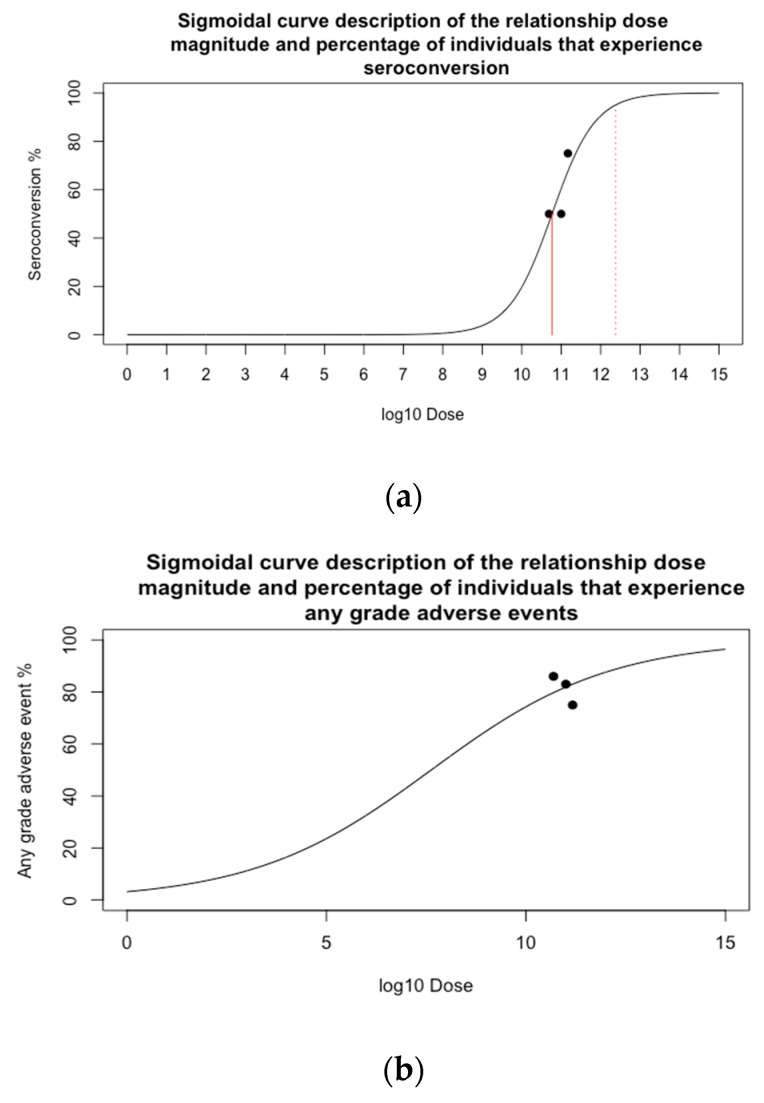
The three curves displaying the relationship between dose and (**a**) percentage of vaccinated individuals predicted to seroconvert, (**b**) percentage of vaccinated individuals predicted to experience any grade adverse events and (**c**) percentage of vaccinated individuals predicted to experience grade 3+ adverse events. The curves are sigmoid curves calibrated to data. Black dots represent the data the curves were calibrated to. In (**a**) the solid and dashed red lines show respectively the doses for which 50% and 90% of individuals are predicted to seroconvert. In (**c**) the solid and dashed red lines show respectively the doses for which 17% and 30% of individuals are predicted to experience grade 3+ adverse events. We note that the percentage of individuals experiencing any grade adverse events in (**b**) qualitatively decreased with increasing dose, whereas the model curve was increasing. This decreasing trend could be explained by the expected stochasticity in the data, hence the sigmoid model did not seem unreasonable (Appendix A).

**Figure 3 vaccines-09-00078-f003:**
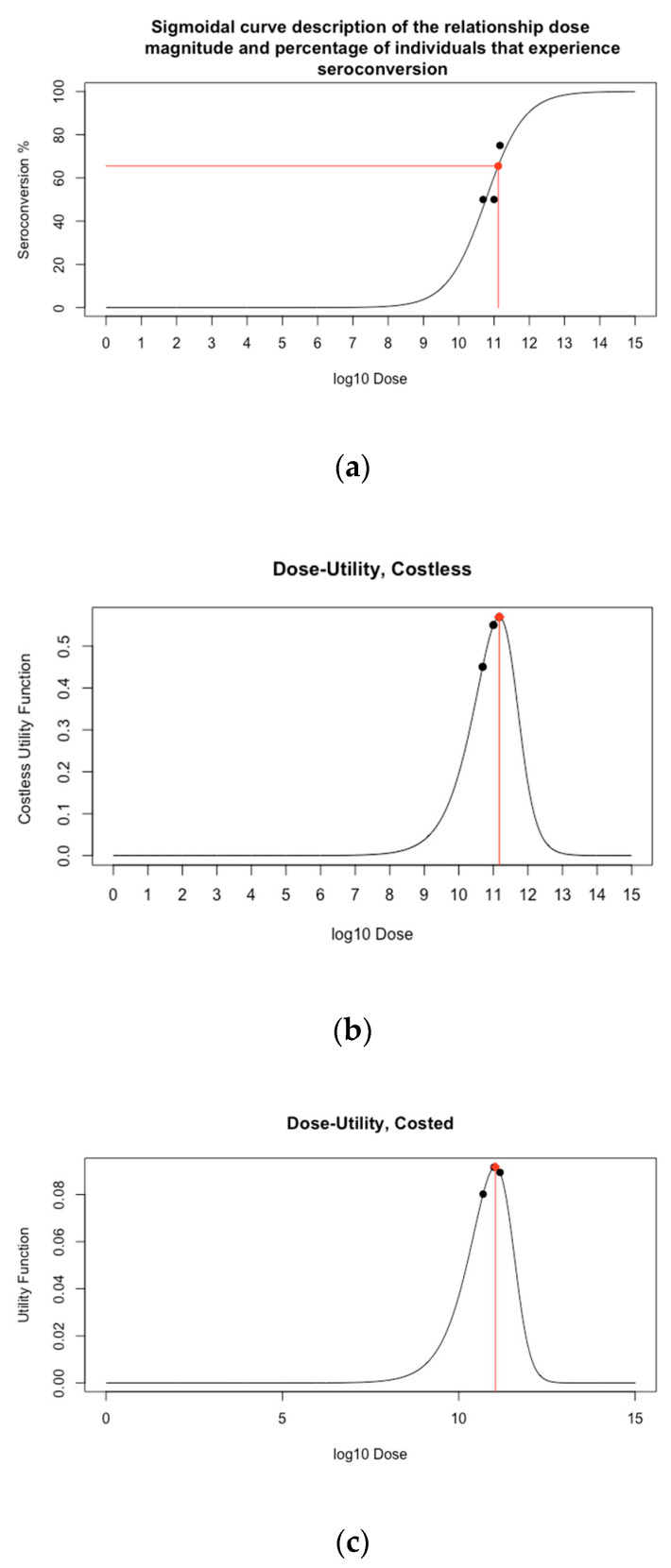
Displays of the predicted utility of doses between 10^0^ and 10^15^ VP. (**a**) shows dose-seroconversion, with the horizontal red line indicating the 65.5% seroconversion threshold required for herd immunity. (**b**) shows the relationship between dose and the costless utility function and (**c**) shows the relationship between dose and the costed utility function. The black dots represent Table 1. 3 × 10^11^, (**b**) 1.5 × 10^11^ VP and (**c**) 1.1 × 10^11^ VP.

**Figure 4 vaccines-09-00078-f004:**
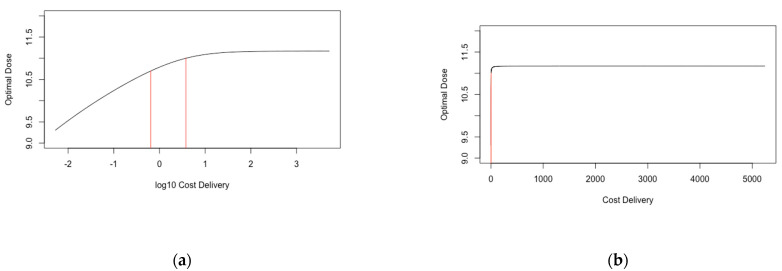
Optimal predicted dose for +/− 3 orders of magnitude around *Cost_Delivery_*. (**a**) has *Cost_Delivery_* at a log10 scale and (**b**) scaled normally. The black line represents the optimal dose, and the red lines indicate the threshold values of *Cost_Delivery_* for which optimal dose is 1 × 10^11^ and 5 × 10^10^ VP.

**Figure 5 vaccines-09-00078-f005:**
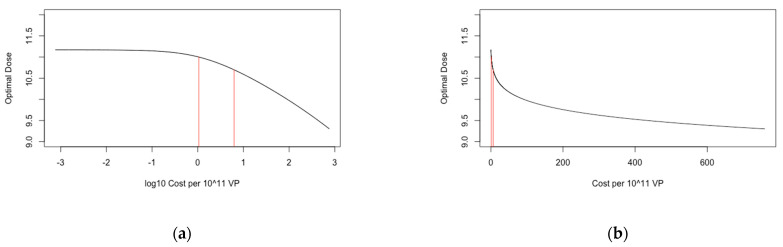
Optimal predicted dose (log10 scale) for +/− 3 orders of magnitude (log10 scale) around *Cost per 10^11^ viral particles*. Table 1. *viral particles* at a log10 scale and the right is scaled normally. The black line represents the optimal dose, and the red lines indicate the threshold values of *Cost per 10^11^ viral particles* for which the optimal dose was 1 × 10^11^ and 5 × 10^10^ VP.

**Table 1 vaccines-09-00078-t001:** Description of adverse event grading in vaccine clinical trials.

Adverse Reaction Grade	General Descriptions
1	Mild. Does not interfere with normal activity
2	Moderate. Interference with normal activity. Little or no treatment required.
3	Severe. Prevents normal activity. Requires treatment.
4	Serious or Potentially Life-Threatening. Generally requires hospitalisation and stopping of any clinical trial where this grade is observed.

**Table 2 vaccines-09-00078-t002:** Parameter values for the cost function. Where needed, a conversion rate of 0.78 U.S. Dollars per GBP was used [26], and a 10-year inflation rate was estimated as 1.35 2020 GBP per 2010 GBP Pound [27].

Name of Parameter	Value	Unit	Description	References
$Cost_{Personnel}\left(£\;per\;vaccination\right)=4.398707$				
AnnualWage	30,615	£ per years	GBP per NHS Band 5 Income per annum (2020/21)	[28]
AnnualHours	1740	hours per years	Work hours per year for average UK nurse	[29]
Timeper vaccination	0.25	hours per vaccination	Recommended hours per vaccination appointment	[30]
$Cost_{Storage}\left(£\;per\;vaccination\right)=0.014$				
$Cost_{Storagepermonth}$	0.014	£ per month	GBP per vaccination per month’s storage. Converted and adjusted for inflation from $0.014 2010 USD.	[31]
$Cost_{materials}\left(£\;per\;vaccination\right)=0.83$				
$Cost_{Gloves}$	0.08	£ per vaccination	GBP of gloves for one vaccination. Converted and adjusted for inflation from $0.08 USD.	[31]
$Cost_{Alcohol}$	0.03	£ per vaccination	GBP of sterilising alcohol for one vaccination. Converted and adjusted for inflation from $0.03 2010 USD.	[31]
$Cost_{PFS}$	0.40	£ per vaccination	GBP of the pre-filled syringe for one vaccination. Converted and adjusted for inflation from $0.39 2010 USD.	[31]
$Cost_{Needles}$	0.32	£ per vaccination	GBP of needle for one vaccination. Converted and adjusted for inflation from $0.31 2010 USD.	[31]
$Costperviralparticle\left(£\;perVP\right)=7.6\times 10^{-12}$				
$Cost_{AdenoviralBatch}$	342,000	£ per Batch	GBP per single-use reference process batch (converted from 450,000 US Dollars)	[32]
Adenoviral Concentration	9 × 10^13^	VPper L	Viral Particles per litre in single-use reference process batch	[32]
Batchvolume	500	L per Batch	Volume of Adenovirus produced in single-use reference process batch	[32]

## Data Availability

Not applicable.

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
