# Peer review of "Optimising Vaccine Dose in Inoculation against SARS-CoV-2, a Multi-Factor Optimisation Modelling Study to Maximise Vaccine Safety and Efficacy"

_vaccines, 2021, doi:10.3390/vaccines9020078_

Round 1
Reviewer 1 Report
To prevent diseases dissemination, vaccination is an appropriate tool which must be balanced with safety, protection and low cost. In this article, authors used modelling and multifactorial optimisation approaches to predict the optimal dose of an adenoviral vectored vaccine against SARS-CoV-2. A dose of 1.1 x 10^11 viral particles was designed to be optimal with respect to seroconversion, safety and cost. These methods could accelerate vaccine development. However, the study has a few shortcomings which are well discussed in the manuscript. This work may be published in this current form.
In my opinion, the content of the article is interesting since using mathematical models to identify the ideal vaccinal doses is an innovative and time-saving strategy. Indeed, the study presents some scientific gaps that the authors were aware as they discuss different points in the manuscript.
But it is clear that considering - a single vaccine campaign with - no placebo group is a big weakness.
In addition, - only 3 points are used to predict the different mathematical curves, and - some of these predictions seem to contradict the dots distribution (e.g., Figure 2B).
- The exact price of viral particles is not known, which makes it difficult to determine the ideal ratio between price and protection. Finally, and this is well discussed in the manuscript, the model was avoiding grade 3+ adverse events that could impair mathematical predictions.
Author Response
To prevent diseases dissemination, vaccination is an appropriate tool which must be balanced with safety, protection and low cost. In this article, authors used modelling and multifactorial optimisation approaches to predict the optimal dose of an adenoviral vectored vaccine against SARS-CoV-2. A dose of 1.1 x 10^11 viral particles was designed to be optimal with respect to seroconversion, safety and cost. These methods could accelerate vaccine development. However, the study has a few shortcomings which are well discussed in the manuscript. This work may be published in this current form.
In my opinion, the content of the article is interesting since using mathematical models to identify the ideal vaccinal doses is an innovative and time-saving strategy. Indeed, the study presents some scientific gaps that the authors were aware as they discuss different points in the manuscript.
We thank the reviewer for their time and consideration in these comments.
But it is clear that considering - a single vaccine campaign with - no placebo group is a big weakness.
We believe that the reviewer makes a great point here. This modelling work does not replace the need for placebo groups to be used in clinical trials, and the absence of such a group forced us to make the model assumption that the base seroconversion rate was 0%. Conducting further work with a placebo group could potentially improve the predictions of the model. Repeating these methods for other vaccine campaigns would also be important future work. We have added this point to the discussion.
4, Lines 444-446: In particular, a placebo group could allow for adjustment of the assumption that individuals that receive no vaccine dose do not seroconvert or experience any adverse events.
In addition, - only 3 points are used to predict the different mathematical curves, and - some of these predictions seem to contradict the dots distribution (e.g., Figure 2B).
3 dosing groups is the minimum amount of data required to calibrate the 3-parameter sigmoid model, and additional data points would improve accuracy in our predictions. However, as data for additional dosing groups were not available, we attempted to account for this as described in supplementary section S2. Having a larger number of dosing groups would have been preferable.
We also agree that the curve in figure 2b does not decrease as dose increases, whereas the data appears to. We note the following considerations.
The data for Figure 2B shows that for the three dosing groups (5.0 x 1010, 1.0 x 1011 and 1.5 x 1011), 86%, 83%, and 75% of individuals experienced any grade adverse events respectively. As the size of each group was 36, we have that 31, 30, and 27 individuals of individuals experienced any grade adverse events respectively. There is a qualitative downwards trend, which our strictly-increasing sigmoid model would be unable to model. However, taking the interpretation that individuals are independent samples of an underlying Bernoulli process we can calculate the 95% confidence interval on the true probability of experiencing any grade adverse events. These are:
-
- For dose 5.0 x1010; 86% (71%,95%)
- For dose 1.0 x 1011; 83% (67%,94%)
- For dose 1.5 x 1011; 75% (58%,88%)
As these confidence intervals do overlap, we did not believe that there was sufficient justification to consider the possibility that an increased dose could reduce the number of adverse events experienced, even given the downward trend observed. We believe it more likely that all three data points have approximately similar probabilities of any grade adverse events.
This is a great point. As this is a trend that may be of note to readers, we have revised response to discuss this trend in the supplementary documents [S6].
- The exact price of viral particles is not known, which makes it difficult to determine the ideal ratio between price and protection.
We agree with the reviewer and the threshold analysis (objective 4, sections 2.5.1 and 3.4.1) highlights this issue.
Finally, and this is well discussed in the manuscript, the model was avoiding grade 3+ adverse events that could impair mathematical predictions.
We thank the reviewer for this comment.
Reviewer 2 Report
The study by Benest et al describes a multifactorial optimization modeling strategy to identify optimal vaccine dosage and thereby maximize vaccine safety and efficacy. The study has significant relevance with regards to developing an in silico method to optimize vaccination safety and efficacy. My minor comments can be found below:
- The authors describe that doses of 5 x 10^10, 1 x 10^11, 1.5 x10^11 VP induced 86%, 83%, 75% any grade adverse events. Can the authors provide an explanation as to why greater percentage of individuals receiving 5 x 10^10 VP experienced more adverse events than individuals receiving 1 x 10^11 VP?
- The authors should provide information on rates of seroconversion with respect to the ages/age groups of the participating cohorts. Accordingly data should also be provided for adverse events corresponding to the age and gender of the individuals.
Author Response
We thank the reviewer for their time and consideration in these comments, and respond below.
The study by Benest et al describes a multifactorial optimization modeling strategy to identify optimal vaccine dosage and thereby maximize vaccine safety and efficacy. The study has significant relevance with regards to developing an in silico method to optimize vaccination safety and efficacy. My minor comments can be found below:
1. The authors describe that doses of 5 x 10^10, 1 x 10^11, 1.5 x10^11 VP induced 86%, 83%, 75% any grade adverse events. Can the authors provide an explanation as to why greater percentage of individuals receiving 5 x 10^10 VP experienced more adverse events than individuals receiving 1 x 10^11 VP?
We agree that the curve in Figure 2b does not appear to well follow the downward trend in the data. We note the following considerations.
The data for Figure 2B shows that for the three dosing groups (5.0 x1010, 1.0 x 1011 and 1.5 x 1011), 86%, 83%, and 75% of individuals experienced any grade adverse events respectively. This represents respectively that for each of the three dosing groups of size N=36 (31,30, and 27) individuals of individuals experienced any grade adverse events. There is a qualitative downwards trend, which our strictly-increasing sigmoid model would be unable to model. However, taking the interpretation that individuals are independent samples of an underlying Bernoulli process we can calculate the 95% confidence interval on the true probability of experiencing any grade adverse events. These are:
-
- For dose 5.0 x1010; 86% (71%,95%)
- For dose 1.0 x 1011; 83% (67%,94%)
- For dose 1.5 x 1011; 75% (58%,88%)
As these confidence intervals do overlap, we did not believe that there was sufficient justification to consider the possibility that an increased dose could reduce the number of adverse events experienced, even given the downward trend observed. We believe it more likely that all three data points have approximately similar probabilities of any grade adverse events.
To illustrate this point please consider the attached plot, which shows the data described. The black line plots the calibrated curve. The red area plots the 95% confidence interval for the percentage of individuals that would experience any-grade adverse events in a 36-per-group trial assuming that this is the true model. As all of the points are within these bounds, again this model seems reasonable with the available data. However, further investigation into the relationship between dose and proportion of individuals experiencing adverse events would be useful if there was sufficient data.
As this is a trend that may be of note to readers, we have adapted this response including the attached plot to discuss this trend in the supplementary documents [S6].
2. The authors should provide information on rates of seroconversion with respect to the ages/age groups of the participating cohorts. Accordingly, data should also be provided for adverse events corresponding to the age and gender of the individuals.
We thank the reviewer for these suggestions. We have included this information in the data section of the results and limitations section of the conclusion, as detailed below.
3.1.1, Lines 262-263
Population demographics including age, gender and pre-existing adenovirus neutralising antibody titre were described [15, Supplementary S5]
4, Lines 425-428
Individuals younger than 45 were shown to be less likely to experience fever and more likely to experience seroconversion [15] which, given that there existed some heterogeneity in age distribution between dosing groups in the data, may impact the model’s future predictive validity.

Reviewer 3 Report
Reviewers comments:
Overview of study: -
John Benest and colleagues reports in this manuscript entitled “Optimizing vaccine dose in inoculation against SARS-CoV-2, a multi-factor optimization modelling study to maximize vaccine safety and efficacy” carried out a mathematical modelling to identify optimal vaccine dosing to minimize cost and increase vaccine efficacy and safety using published human phase 1 clinical trial data (“Safety, tolerability, and immunogenicity of a recombinant adenovirus type-5 vectored COVID-19 vaccine: a dose-escalation, open-label, non-randomised, first-in-human trial”). This was a Ad5 vectored COVID-19 vaccine study conducted in Wuhan, China and it was a dose-escalation, single-centre, open-label, non-randomized, phase 1 trial. Authors have used the data originated from 108 participants, age ranged between 18 and 60 years with mean average age 36.3 years. This is a good data set as it has immune responses (humoral and adaptive) and safety profiles generated from three different doses low dose 5 × 10¹⁰ (n=36), middle dose 1 × 10¹¹ (n=36), or high dose 1·5 × 10¹¹ (n=36) of the Ad5 vectored COVID-19 vaccine. This data set was used to quantitate following three important aspect of a vaccine- the minimum dose that may induce herd immunity, the dose that maximizes immunogenicity and safety and the dose that maximizes immunogenicity and safety whilst minimizing cost.
In conclusion, the authors were able draw some important conclusion from their modeling analysis including, dose-safety relationship, does-seroconversion relationship, identify the minimum dose that is predicted to theoretically induce herd immunity, identify the dose that maximizes immunogenicity and safety by keeping cost in attention. However, there are important parameters like how seroprevalence of Ad-based vectors, infection rates influence achieving herd immunity in combination with vaccines is not discussed anywhere in this context of mathematical modeling. Addressing durability of single dose Ad5-vecotered COVID-19 vaccine responses and two-dose Ad5 vaccination regimens and comparing with cost and timelines would layout for COVID-19 vaccine development. If possible, authors should fit the data from ChAdOx1 nCoV-19 vaccine studies and strengthen this study by proving the same aspects.
In conclusion, I believe this modeling study has drawn important conclusions from clinical trial data set, laying foundations to develop further models in understanding dynamics and development of new vaccine strategies by keeping dose and cost in the context. Therefore, it is worth publishing in Vaccines journal-MDPI with minor revision.
Author Response
We thank the reviewer for their time and consideration in these comments, and respond below.
John Benest and colleagues reports in this manuscript entitled “Optimizing vaccine dose in inoculation against SARS-CoV-2, a multi-factor optimization modelling study to maximize vaccine safety and efficacy” carried out a mathematical modelling to identify optimal vaccine dosing to minimize cost and increase vaccine efficacy and safety using published human phase 1 clinical trial data (“Safety, tolerability, and immunogenicity of a recombinant adenovirus type-5 vectored COVID-19 vaccine: a dose-escalation, open-label, non-randomised, first-in-human trial”). This was a Ad5 vectored COVID-19 vaccine study conducted in Wuhan, China and it was a dose-escalation, single-centre, open-label, non-randomized, phase 1 trial. Authors have used the data originated from 108 participants, age ranged between 18 and 60 years with mean average age 36.3 years. This is a good data set as it has immune responses (humoral and adaptive) and safety profiles generated from three different doses low dose 5 × 10¹⁰ (n=36), middle dose 1 × 10¹¹ (n=36), or high dose 1·5 × 10¹¹ (n=36) of the Ad5 vectored COVID-19 vaccine. This data set was used to quantitate following three important aspect of a vaccine- the minimum dose that may induce herd immunity, the dose that maximizes immunogenicity and safety and the dose that maximizes immunogenicity and safety whilst minimizing cost.
In conclusion, the authors were able draw some important conclusion from their modeling analysis including, dose-safety relationship, does-seroconversion relationship, identify the minimum dose that is predicted to theoretically induce herd immunity, identify the dose that maximizes immunogenicity and safety by keeping cost in attention. However, there are important parameters like how seroprevalence of Ad-based vectors, infection rates influence achieving herd immunity in combination with vaccines is not discussed anywhere in this context of mathematical modeling. Addressing durability of single dose Ad5-vecotered COVID-19 vaccine responses and two-dose Ad5 vaccination regimens and comparing with cost and timelines would layout for COVID-19 vaccine development. If possible, authors should fit the data from ChAdOx1 nCoV-19 vaccine studies and strengthen this study by proving the same aspects.
Thank you. We agree these suggestions are important considerations from the perspectives of both adenoviral vector vaccine development and modelling.
We had added the following text to the body of the work in relation to comparing it with expected cost.
3.4., Lines 312-313
The predicted cost is within the expected range [$5-$37] for a single SARS-CoV-2 vaccine dose [34].
We have added the following text to the body of the work in relation to the potential impact of adenoviral seroprevalence:
3.1.1., Lines 262-263
Population demographics including age, gender and pre-existing adenovirus neutralising antibody titre were described [15, Supplementary S5]
4., Lines 424-425
Additionally, prior adenoviral exposure or age of vaccinated individuals could impact the probabilities of seroconversion and adverse events.
4., Lines 457-459
Given sufficient data, incorporating the covariates of age and pre-existing rAd5 neutralising antibody titre into the model could aid in predictive validity across various populations and in assessing dose effect.
We have added the following text to the body of the work in relation to the potential impact population parameters, boost doses and the potential importance of response at different time points.
4., Lines 421-424
Further limitations are due to the non-inclusion of potential population effects and covariates in the model. The proportion of individuals that the vaccine needs to protect may change depending on the number of individuals that have been previously infected or on the extent that a prior infection provides lasting immunity.
4., Lines 460-462
Sixthly, this work only considers a single dose of the vaccine and response at one time-point. Further modelling should be attempted to address dose-optimisation at different time points or to consider a prime-boost paradigm.
We agree with the reviewer that the ChAdOx1 nCoV-19 vaccine would be an ideal vaccine candidate to consider for dose optimisation. However, there is an insufficient number of dosing levels to apply our methods.
In conclusion, I believe this modelling study has drawn important conclusions from clinical trial data set, laying foundations to develop further models in understanding dynamics and development of new vaccine strategies by keeping dose and cost in the context. Therefore, it is worth publishing in Vaccines journal-MDPI with minor revision.
We again thank the reviewer for their useful comments that have improved the manuscript.
Round 2
Reviewer 2 Report
The authors have addressed the concerns raised.
A minor suggestion: the authors should consider moving the data and information in supplementary document S6 to address concern #1 to the main figure/text.
Author Response
We thank the reviewer for their suggestion. We have added the following text to the description of Figure 2b to increase clarity in the discussion of the qualitative downward trend.
We note that the percentage of individuals experiencing any grade adverse events in (b) qualitatively decreased with increasing dose, whereas the model curve was increasing. This decreasing trend could be explained by the expected stochasticity in the data, hence the sigmoid model did not seem unreasonable [Supplementary 6].
We again thank the reviewer for both rounds of their comments, which have improved the manuscript.